# Diagnostic Accuracy of Various Radiological Measurements in the Evaluation and Differentiation of Flatfoot: A Cross-Sectional Study

**DOI:** 10.3390/diagnostics12102288

**Published:** 2022-09-22

**Authors:** Fayaz Khan, Mohamed Faisal Chevidikunnan, Mashael Ghazi Alsobhi, Israa Anees Ibrahim Ahmed, Nada Saleh Al-Lehidan, Mohd Rehan, Hashim Abdullah Alalawi, Ahmed H. Abduljabbar

**Affiliations:** 1Department of Physical Therapy, Faculty of Medical Rehabilitation Sciences, King Abdulaziz University, Jeddah 22252, Saudi Arabia; 2King Fahd Medical Research Center, King Abdulaziz University, Jeddah 22252, Saudi Arabia; 3Department of Medical Laboratory Sciences, Faculty of Applied Medical Sciences, King Abdulaziz University, Jeddah 22252, Saudi Arabia; 4Department of Radiology, Faculty of Applied Medical Sciences, King Abdulaziz University, Jeddah 22252, Saudi Arabia; 5Department of Radiology, Faculty of Medicine, King Abdulaziz University Hospital, Jeddah 22252, Saudi Arabia

**Keywords:** flatfoot, X-rays, pes planus, diagnosis, evaluation

## Abstract

Arch angle is used to indicate flatfoot, but in some cases, it is not easily defined. The presence of flatfoot deformity remains difficult to diagnose due to a lack of reliable radiographic assessment tools. Although various assessment methods for flatfoot have been proposed, there is insufficient evidence to prove the diagnostic accuracy of the various tools. The main purpose of the study was to determine the best radiographic measures for flatfoot concerning the arch angle. Fifty-two feet radiographs from thirty-two healthy young females were obtained. Five angles and one index were measured using weight-bearing lateral radiographs; including arch angle, calcaneal pitch (CP), talar-first metatarsal angle (TFM), lateral talar angle (LTA), talar inclination angle (TIA) and navicular index (NI). Receiver-operating characteristics were generated to evaluate the flatfoot diagnostic accuracy for all radiographic indicators and Matthews correlation coefficient was calculated to determine the cutoff value for each measure. The strongest correlation was between arch angle and CP angle [r = −0.91, *p* ≤ 0.0001, 95% confidence interval (CI) (from −0.94 to −0.84)]. Also, significant correlations were found between arch angle and NI [r = 0.62, *p* ≤ 0.0001, 95% CI (0.42 to 0.76)], and TFM [r = 0.50, *p* ≤ 0.0001, 95% CI (from 0.266 to 0.68)]. Furthermore, CP (cutoff, 12.40) had the highest accuracy level with value of 100% sensitivity and specificity followed by NI, having 82% sensitivity and 89% specificity for the cutoff value of 9.90. In conclusion, CP angle is inversely correlated with arch angle and considered a significant indicator of flatfoot. Also, the NI is easy to define radiographically and could be used to differentiate flat from normal arched foot among young adults.

## 1. Introduction

Flatfoot is a common foot pathology characterized by a collapse of the medial longitudinal arch (MLA) of the foot. As a result, the entire sole of the foot makes total contact with the ground [1]. Additionally, it is characterized by the abduction of the forefoot and eversion of the calcaneus [2]. The MLA has a concave shape that acts as a shock absorber during walking, jumping, and running on different surfaces [3]. The prevalence of flatfoot was reported to be approximately 3–10% among the adult population [4]. Among adults, flatfoot can be asymptomatic or symptomatic, resulting in various clinical consequences [5]. Flatfoot may lead to many deformities or disabilities if it remains undiagnosed [6], and further consequences may include pain in the lower extremities [7], back pain [8], stress fractures [9], and hallux valgus deformities [10].

The foot is an important component of the body that maintains adequate balance when people move in their gaits. Individuals with foot deformities are more susceptible to falls and loss of balance, which affect their quality of life (QoL) [11,12,13]. In clinical practice, foot structural deformities have a high prevalence of 50% to 80% among adults [14]. Several studies reported the negative impacts of different foot deformities on the QoL. A recent study conducted by López-López et al. (2021) [15] investigated the relationship between QoL and foot health among individuals with and without foot pathologies. The researchers stated that foot disorders negatively affect people’s daily life activities. Additionally, another study highlighted the importance of considering the relationship between QoL and foot structure deformities [16]. In that context, the present study was conducted to discover the best radiographic measure for detecting flatfoot, which may assist in improving the quality of care among young adults.

Clinically, there are several approaches to identifying flatfoot deformity, including clinical diagnosis [17], footprint, and radiologic assessment [18]. In radiographic assessment, the weight-bearing MLA has been utilized as a diagnostic assessment tool to determine the presence of flatfoot [19]. While it is considered the standard diagnostic method, numerous studies have found discrepancies in MLA radiographic measurements among subjects [20,21,22,23]. Many radiological measurements can be determined from the weight-bearing X-ray [24]. A set of radiological measurements can be defined from a posterior or lateral radiographic view of the weight-bearing foot to identify flatfoot deformity. The arch angle and calcaneal pitch angle (CP) have been used radiographically to evaluate the MLA [25]. In a study conducted in Taiwan, military recruits were determined to have flatfoot if the arch angle was ≥165°. However, researchers revealed that the assessment of arch angle was restricted because the image quality was affected by the superimposition of the metatarsal bones [6]. It was recommended that the CP angle should be measured to distinguish flatfoot from a normal arch when it is challenging to define the arch angle.

Today, several radiographic assessment techniques are proposed for use in the MLA assessment, such as: talar-first metatarsal angle (TFM), lateral talar angle (LTA), talar inclination angle (TIA), and navicular index (NI) [22,23,26,27], however none of these measures are universally agreed on. In this study, we aimed to find the best radiographic measurement for diagnosing flatfoot among healthy females. According to the available studies, there is limited literature regarding the measurement of flatfoot angles among healthy, young females. This study was conducted to provide information about cutoff values for the five radiographic measures (CP, TFM, LTA, TIA, and NI), to facilitate the improved interpretation and evaluation of flatfoot radiographs. Additionally, we aimed to identify the relationship between these sets of radiographic measures by keeping the arch angle as the reference value. This study hypothesized that radiographic measures represent simple and easy tools to use to diagnose flatfoot with a high level of sensitivity and specificity. Furthermore, the researchers of this study hypothesized that relationships exist between the five radiographic measures (CP, TFM, LTA, TIA, and NI) and the arch angle.

## 2. Materials and Methods

### 2.1. Subjects and Setting

Researchers of this study used the Foot Posture Index (FPI ≥ +6) to screen for the flatfoot. Subjects who scored +6 or above on the FPI for one or both feet were included in the study. Subjects who had any history of surgery or restricted foot and/or ankle range of motion were excluded from the study. After fulfilling the criteria, only thirty-two healthy young females aged between 18 to 25 years were recruited for the study. The minimum sample size was calculated by using G-power 3.1 software (G-power v3.1. https://gpower.software.informer.com/3.1/ (accessed on 28 March 2022)) to achieve a power of 0.80. In G-power, a correlation test was selected for a priori power calculation with a medium effect size of 0.7 and significance level of 0.05. Thirty-nine feet were estimated to be the minimum sample needed to reach a power of 0.8 and in the current study fifty two feet were included for the analysis. This cross-sectional study was conducted in the Department of Physical Therapy, King Abdulaziz University. The report of this study has been written according to the Standards for Reporting Diagnostic accuracy studies (STARD) [28] and the Strengthening the Reporting of Observational Studies in Epidemiology (STROBE) guidelines (Appendix A) [29].

### 2.2. Procedure

Ethical approval was obtained from the Center of Excellence in Genomic Medical Research (number: 05-CEGMR-Bioeth-2019), approved by the National Committee of Bioethics (KACST: HA-02-J-003). Also, a written informed consent was obtained from each subject before they were included in the study. Demographic characteristics of the subjects including age, gender, BMI, and FPI-6 were documented for each of the subjects. Then, subjects were asked to take off their shoes and socks for barefoot examination to determine subjects’ eligibility for the study. FPI-6 is considered a good reliable clinical flatfoot measurement tool with inter-rater reliability between 0.62 to 0.91 and intra-rater reliability of 0.81 to 0.91 [30], and it is used to determine the presence of flatfoot in either right or left foot. The first author F.K, who is a senior author with 15 years of experience, screened the subjects. Also, an experienced radiologist who was blinded to the study took the foot X-rays. The first and second author M.C, who has 20 years of experience in the musculoskeletal field, measured the angles using RadiAnt DICOM software (RadiAnt DICOM v4.2. https://www.radiantviewer.com/ (accessed on 12 April 2022)).

Figure 1 shows a flowchart of the subject selection process. All subjects were instructed to walk a few steps, then stand in a relaxed–static standing position with head in a neutral position and both arms by their sides. The six components of the FPI-6 are: (a) palpation of the talonavicular head, (b) observation and comparison of the superior and inferior lateral malleolus curves, (c) observation of the inversion and eversion of the calcaneus in the frontal plane, (d) protrusions in the region of the talonavicular joint, (e) height and congruence of the medial longitudinal arch, and (f) abduction/adduction of the forefoot on the rearfoot. Each component of the six observations was measured and graded as 0 for a neutral foot position, at least −2 for a clear indication of foot supination, and at most +2 for a clear indication of foot pronation. The total FPI score for all components was between −12 and +12. Foot posture was classified as normal if the total score was between 0 and +5, supinated if the score was between −1 and −12, or pronated if it was scored from +6 to +12. Only subjects who scored +6 and above were included in the study [31,32].

After determining the eligibility of the subjects, foot X-rays were taken by an experienced radiologist who was blinded to the study. A lateral weight-bearing radiograph was taken for each foot separately while the subject was standing straight (in an upright neutral position) on a table and the other foot was raised. The X-ray system used for the study was a DR Definum 6000 machine (General Electric Company, Boston, USA) with a 17 × 14-inch cassette. The placement of the cassette was between the medial borders of the hindfoot, maintained vertically in the groove. From a fixed distance of 100 cm, the X-ray tube was directed vertically toward the cassette. The exposure was set at 52 kV and 4.5 mA for the lateral projection. The central X-ray beam was aimed toward the navicular bone.

In our study, five angles and one index were measured to determine flatfoot on the radiographs, including the arch angle, CP, TFM, LTA, TIA, and NI. The arch angle was used as the reference standard for all other radiographic measures in this study. The arch angle was measured at the intersection of two lines: the calcaneal line (a line drawn along the inferior surface of the calcaneus) and the fifth metatarsal line (a line drawn along the inferior edge of the fifth metatarsal bone) [33]. Subjects were determined to have flatfoot if the arch angle was ≥165° [34]. The CP angle was drawn by a horizontal line (a line drawn horizontally from calcaneus to the inferior surface of the 5th metatarsal head inferior surface) and the calcaneal line [33]. The TFM angle was made by the intersection of the two longitudinal axes of the first metatarsal and talus. The talus longitudinal axis was the line connecting the centers of the talar head and neck parts in its narrowest width [35,36]. The lateral talar angle was the angle created between the talus line, which runs through the center point of the body and neck of the talus, and the calcaneal line. The talar inclination angle was made between the horizontal line and the talar line [22]. To determine the NI, the longitudinal arch length was divided by the navicular height measured from the floor [26] (Figure 2a,b).

### 2.3. Statistical Analysis

The data were analyzed using the Statistical Package for Social Sciences version 21.0 (Statistical Package for Social Sciences v21.0. https://www.ibm.com/support/pages/downloading-ibm-spss-statistics-21 (accessed on 24 May 2022) and GraphPad Prism (GraphPad Prism v7.0. https://www.graphpad.com/guides/prism/7/user-guide/ (accessed on 18 June 2022)). Descriptive statistics were used to describe the demographic characteristics of the sample. Mean, median and standard deviation were calculated for age, BMI, FPI and all radiographic measurements. Correlation analyses were used to identify the associations between arch angle and the five radiographic measurements. Correlation coefficients (r) were classified as follows: little or no association (r = 0–0.24), fair (r = 0.25–0.49), moderate–good (r = 0.50–0.74), and good–excellent association (r = 0.75–1) [37]. Before conducting the correlation analysis, data were checked for normality to perform the suitable test by conducting the Shapiro–Wilk test. The receiver-operating characteristic (ROC) test was conducted for the CP, TFM, LTA, TIA and NI measures, compared to the arch angle for predicting flatfoot. Matthews correlation coefficient (MCC) was used to define the cutoff value for all flatfoot radiographic measurements. A *p*-value of 0.05 or less was considered statistically significant for all analyses.

### 2.4. Receiver-Operating Characteristic (ROC) Curve

The ROC test is a popular and widely used method to evaluate the performance of a binary classifier model. The ROC curve is generated by plotting the true positive rate (TPR) versus the false positive rate (FPR) with various cutoff settings for the binary classifier. The area under the curve (AUC) or ROC space provides a measure of the effectiveness of the binary classifier. An ideal classifier will cover 100% area (AUC = 1.0) and a random classifier will cover 50% area (AUC = 0.5) with all the points along the diagonal.

### 2.5. Matthews Correlation Coefficient (MCC)

The MCC or phi coefficient is a measure of the quality of a binary classifier, calculated as follows:$$\mathrm{MCC}=\frac{TP\times TN-FP\times FN}{\sqrt{\left(TP+FP\right)\left(TP+FN\right)\left(TN+FP\right)\left(TN+FN\right)}}$$

The value of MCC ranges between −1 and +1. The value of MCC + 1 represents an ideal classifier, 0 a random classifier, and −1 a total disagreement between prediction and observation. The maximum value of MCC was used to determine the cutoff value for each classifier.

## 3. Results

The radiographic data of fifty-two feet images were included in the analysis. The mean age of the subjects was 20.69 ± 1.15 years (range = 18–25 years) and their mean BMI was 23.02 ± 3.79 kg/m² (range = 16.20–33.30). The mean values of the five radiographic angles and the NI are illustrated in Table 1.

All radiographic measurements were significantly correlated with the arch angle (*p* ≤ 0.05). All the angles and NI had a positive correlation with the arch angle, except for the CP angle, which had a negative relationship. We found the strongest correlation was between the arch angle and the CP angle (r = −0.91, *p* ≤ 0.0001, 95% CI (from 0.94 to −0.84)). Additionally, a significant relationship was found between the arch angle and the NI (r = 0.62, *p* ≤ 0.0001, 95% CI (from 0.42 to 0.76)), and the TFM (r = 0.50, *p* ≤ 0.0001, 95% CI (from 0.27 to 0.68)). However, we found a weak relationship was found between the arch angle and the LTA (r = −0.49, *p* = 0.0002, 95% CI (from −0.67 to −0.24)), and the TIA (r = 0.32, *p* = 0.021, 95% CI (from 0.05 to 0.55)) (Figure 3).

The ROC test showed that the CP angle and NI were perfect classifiers for flatfoot, with AUCs of 1 and 0.9, respectively (Figure 4 and Table 2). We found a CP angle cutoff of 12.40 yielded high accuracy, with a sensitivity and specificity of 1 for the flatfoot diagnosis, while NI with a cutoff value of 9.90 yielded 0.82 sensitivity and 0.89 specificity. Meanwhile, the LTA (0.76 sensitivity and 0.83 specificity) had a cutoff value of 41.8 and MCC of 0.58, while the TFM angle (0.65 sensitivity and 0.86 specificity) had a cutoff value of 13.4 and MCC of 0.51. Similarly, the TIA angle (0.88 sensitivity and 0.37 specificity) had a cutoff value of 24.60 and MCC of 0.26 (Table 2).

## 4. Discussion

The aim of this study was to determine the best radiographic measures for diagnosing flatfoot concerning arch angle measurements. Although numerous flatfoot diagnostic procedures have been proposed in the literature, there remains no standard diagnostic measure to determine the presence of flatfoot. According to the available studies, the accuracy levels of these flatfoot diagnostic measures have not yet been determined [25]. In our study, we discovered that the CP angle was the best flatfoot indicator among all radiographic measures, which was indicated by its perfect sensitivity and specificity, as well as its PPV and NPV values. In addition, the CP angle had a strong negative correlation with the arch angle, and it had an AUC value equal to 1. Our findings were consistent with the results of Huan-Chu Lo et al., who found that the CP angle was a significant indicator of flatfoot. Our findings further suggested that the CP angle might be the best radiographic measure to predict flatfoot after the arch angle. In our study, the CP angle cutoff of <12.40° was determined to identify flatfoot, with a sensitivity and specificity of 1, which was in line with the results of Huan-Chu Lo et al., who determined the cutoff value for the CP angle to be <12.30°. The CP angle represents a useful indicator to distinguish a normal foot from flatfoot because it can be defined easily on the foot radiograph by the intersection of the calcaneal and horizontal lines [6].

An important finding of this study was the significant association between the NI and arch angle. The NI had a cutoff value of 9.90, with 0.82 sensitivity and 0.89 specificity. To the best of our knowledge, no previous studies assessed the NI as a diagnostic measure for adult flatfoot. However, a study was conducted by Roth et al. on children to identify the association of NI with other flatfoot diagnostic measurement angles, and they stated that a NI cutoff value of 6.74 distinguished flatfeet from normal feet among children, with 0.86 sensitivity and 0.75 specificity, which was similar to our findings [26]. Amongst the different flatfoot measuring angles described in the literature, the NI represents an easy and quick method to determine flatfoot because the determination procedure does not require high-quality X-ray images; instead, the NI can simply be obtained by dividing the height of the longitudinal arch by the navicular bone height. Therefore, the NI might offer a valuable radiographic measuring tool to determine flatfoot among young adults.

On the lateral radiographic view, the TFM angle was determined by measuring the talar inclination and heel pronation, and it was moderately associated with the arch angle. In this study, the TFM cutoff value was found to be 13.4°, with 0.65 sensitivity and 0.86 specificity, which was a slightly higher cutoff value than that found previously in the literature (9.58, with a high specificity of 0.90) [6]. The cutoff value discrepancy might be attributable to the different flatfoot screening criteria used, i.e., FPI ≥ 6 in our study vs. arch angle ≥165° in the study of Huan-Chu Lo et al. [6].

In contrast to all other radiographical measures, the LTA and TIA were weakly correlated with the arch angle reference. Additionally, the AUC values for both angles were below 0.9, signaling they were inefficient indicators of flatfoot, in keeping with the previous findings [20,23,27]. The LTA was not easily determined because of the irregular shape of the calcaneus and talus. This was highlighted in a recent study by Hamel et al., who noted the difficulty of defining this angle on foot radiographs [27].

It is not easy to identify adult flatfoot and there are no definitive diagnostic tools for that purpose. The results of this study defined several radiographic measures that can be used in clinical settings. A key strength of this study was discovering that the NI offers an easy and precise radiographic measure for flatfoot thanks to the simplicity of its measurement, i.e., dividing the length of the foot arch by the navicular height. Yet, as with any research, there were some limitations to the findings of this study. Our sample was limited to healthy, young females, which may limit the applicability of the findings to other populations. Future studies may compare males and females to assess and confirm the results of this study. Additionally, further research could include a wide range of ages among both genders, which is recommended to confirm the results presented in this report. Furthermore, our study was limited to the radiographic assessment methods; to go beyond this, future studies may compare the radiographic measures with other flatfoot diagnostic methods such as footprint analysis. Additionally, this study failed to investigate whether there was a significant difference in radiographic or demographic features of adult flatfoot, which could be investigated in the future.

In conclusion, the study findings suggest that the CP angle and the NI can be used as indicators to determine the presence of flatfoot. Moreover, the study findings demonstrate strong correlations between the arch angle, CP angle, TFM angle, and NI. The CP angle and NI may represent the best radiographic measuring tools to evaluate the presence of flatfoot among young adults. The results of this study produce a baseline for the radiographic measures that can be used to indicate flatfoot. The CP angle and NI are simple and accurate identifiers of flatfoot that can be easily applied in clinical settings. It is essential to highlight the importance of determining flatfoot because of its association with fall risk and balance problems among adults.

## Figures and Tables

**Figure 1 diagnostics-12-02288-f001:**
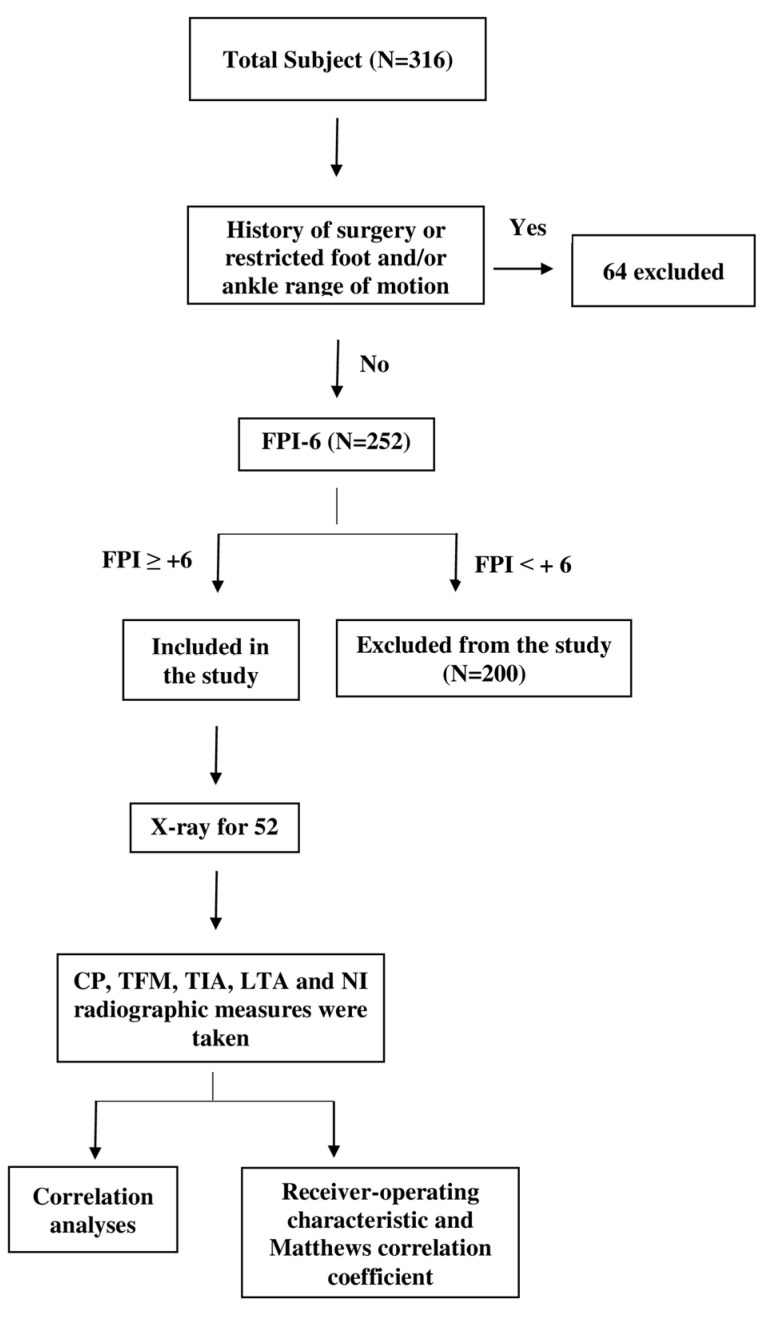
Flowchart of the study selection procedure and design.

**Figure 2 diagnostics-12-02288-f002:**
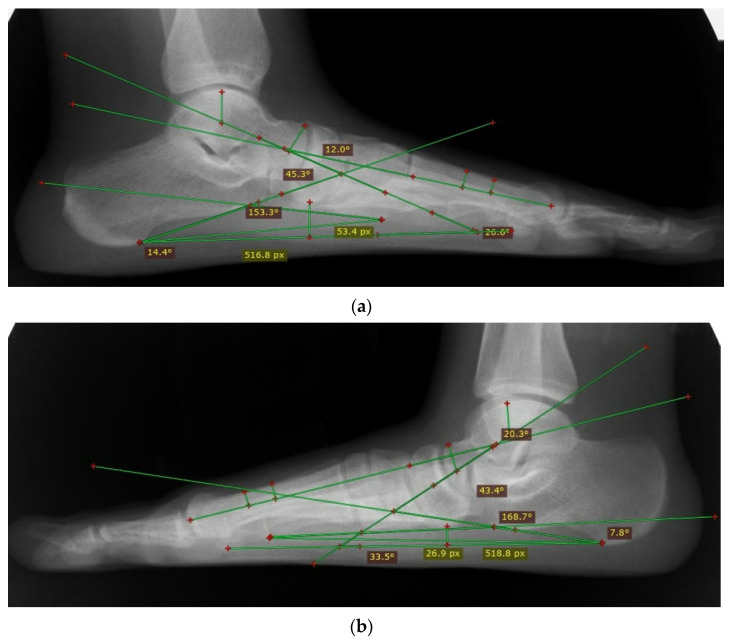
(**a**): Lateral radiographic assessment of a normal weight-bearing foot demonstrating the arch angle = 153.3°, CP angle = 26.6°, TFM = 12°, TIA = 14.4°, LTA = 45.3°, and NI = 9.68. (**b**): Lateral radiographic assessment of a flatfoot weight-bearing foot demonstrating the arch angle = 168.7°, CP angle = 7.8°, TFM = 20.3°, TIA = 33.5°, LTA = 43.4°, and NI =19.5.

**Figure 3 diagnostics-12-02288-f003:**
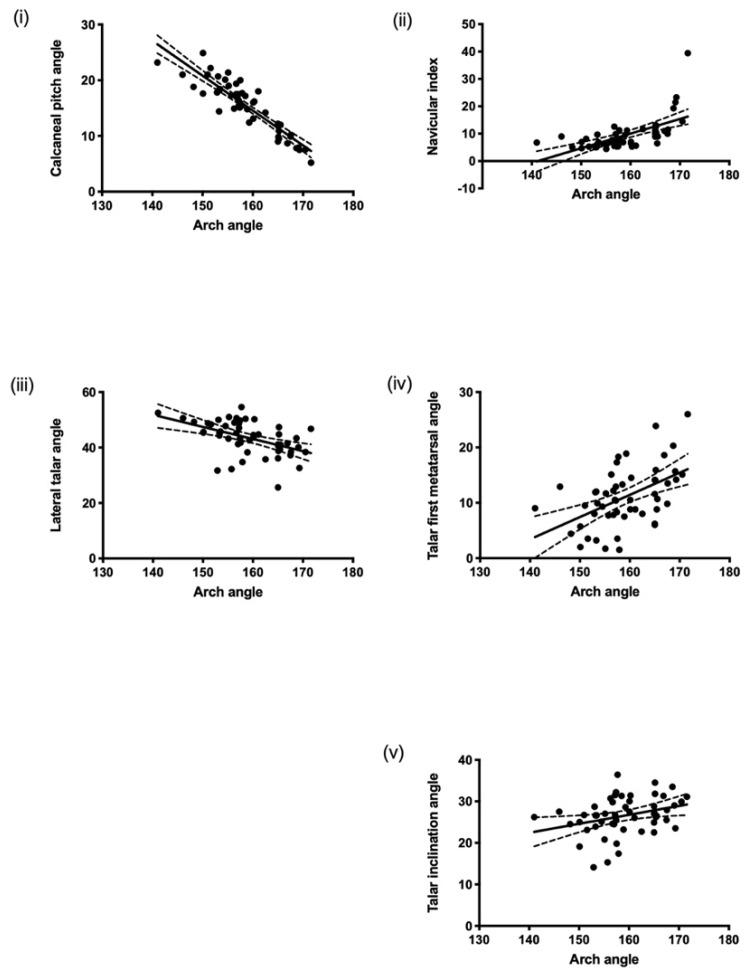
Correlation of arch angle with (**i**) calcaneal pitch angle, (**ii**) navicular index, (**iii**) lateral talar angle, (**iv**) talar-first metatarsal angle, and (**v**) talar inclination angle.

**Figure 4 diagnostics-12-02288-f004:**
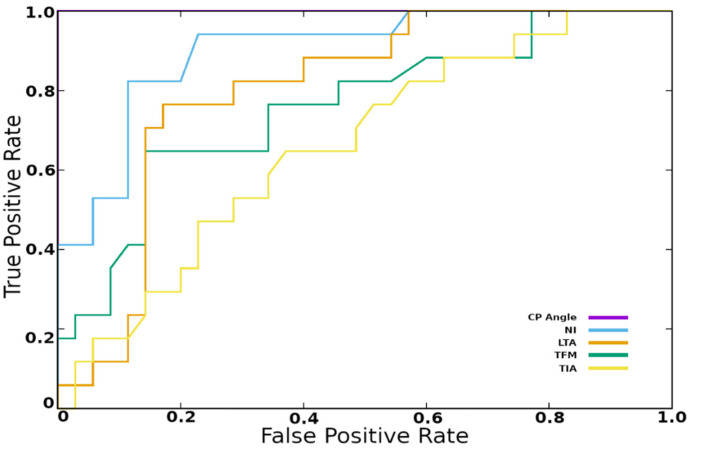
Receiver-operating characteristic (ROC) curves demonstrating the sensitivity (true positive rate) and specificity (1—false positive rate) of the arch angle, calcaneal pitch angle (CP), talar-first metatarsal angle (TFM), talar inclination angle (TIA), and lateral talar angle (LTA) when used for predicting flatfoot.

**Table 1 diagnostics-12-02288-t001:** Demographic characteristics and baseline scores (*n* = 52).

Variables	Mean ± SD	Median (Range)	95% CI
Age (years)	20.69 ± 1.15	21 (18–25)	20.37–21.01
BMI (kg/m²)	23.02 ± 3.79	23.01 (16.20–33.30)	21.96–24.08
FPI-6	7.22 ± 2.76	9 (6–11)	6.45–7.99
Arch angle	159.1 ± 6.74	157.8 (141–171.6)	157.2–161
CP	15.14 ± 4.66	15.70 (5.2–24.9)	13.85–16.44
TFMA	11.06 ± 5.37	10.50 (1.5–26.00)	9.56–12.55
LTA	43.50 ± 6.04	44.45 (25.60–54.60)	25.29–27.82
TIA	26.56 ± 4.54	26.65 (14.10–36.40)	25.29–27.82
NI	9.54 ± 5.82	8.03 (4.42–39.44)	7.92–11.16

SD: standard deviation; 95% CI: confidence interval.

**Table 2 diagnostics-12-02288-t002:** ROC measures including AUC or ROC space and MCC cutoff values for the four radiographic parameters. The MCC cutoff column shows the cutoff values, with the MCC values in parentheses.

Parameter	AUC	MCC Cutoff	FPR (1-Specificity)	TPR (Sensitivity)	PPV	NPV
CP	1.00	12.40 (1.00)	0	1	1	1
NI	0.90	9.90 (0.70)	0.11	0.82	0.78	0.91
LTA	0.8	41.8 (0.58)	0.17	0.76	0.68	0.88
TFM	0.76	13.4 (0.51)	0.14	0.65	0.69	0.83
TIA	0.66	24.60 (0.26)	0.63	0.88	0.40	0.87

## Data Availability

The data presented in this study are available on request from the corresponding author. The data are not publicly available due to privacy restrictions.

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
