# Peer review of "Diagnostic Accuracy of Various Radiological Measurements in the Evaluation and Differentiation of Flatfoot: A Cross-Sectional Study"

_diagnostics, 2022, doi:10.3390/diagnostics12102288_

Round 1

Reviewer 1 Report

I congratulate the authors on an ambitious cross-sectional study. The research is robust and the design well considered. I look forward to seeing the end result of this work when it is finally complete and published. I commend the authors for their work - both all of the work leading up to this point and for the planning of this investigation - their contribution to the low back pain and flat feet literature. I do have some comments about certain methodological issues covered below under MAJOR ISSUES the majority of which are related to clarity of the work as it is currently written.

TITLE

The title should be amended slightly to ensure that the reader understands the type of research immediately that this paper for clarity, interes and ease of read.

ABSTRACT

It is hard to get the detail in an abstract when the word count is limited and this is often the hardest part of a paper to write. However, I do feel that it would be beneficial to explain what specifically you are looking at in relation to assesment and flat feet (this also applies to the main body of the paper). Is it the development of flat feet associated with poor diagnostic . This needs to be made clearer throughout the paper

KEYWORDS:

Please use recognised MeSH terms as this will assist others when they are searching for information on your research topic. The following website will provide these (simply start typing in a keyword and see if it exists or find an alternative if it does not): https://www.ncbi.nlm.nih.gov/mesh

The introduction is weak. An introduction should announce your topic, provide context and a rationale for your work, while catching the reader´s interest and attention. The above has not been given in the introduction that I have read. Thus, I suggest in this section should be improved, with more details about prevalence, impact related with flat feet associated with the impact of the quality of health. It is indeed important paper but it lacks several critical references, in which it was presented related with this condition, and it should be emphasized in the INTRODUCTION or Discussion of the authors' paper. More info info in the research of Calvo et al entitled Foot Arch Height and Quality of Life in Adults https://pubmed.ncbi.nlm.nih.gov/30041462/. Furthermore, in relation with foot problems and flat feet to revise the researcher of Navarro et al entitled Impact of quality of life related to foot problems: a case-control study https://pubmed.ncbi.nlm.nih.gov/34267276/

Also, please describe the hypothesis in this section.

MATERIAL AND METHODS:

This section are appropriate and described in adequate detail while the conclusions clearly link to the data presented. Please, expand and clarification information related with this research for adhere to reporting STROBE guidelines.

RESULTS:

The results section is very appropriate according to the developed methods and the journal´s scope.

DISCUSSION:

Include this section the principal strengths and weaknesses in relation to other studies, discussing important differences in results; the meaning of the study: possible explanations and implications and unanswered questions and future research

CONCLUSION:

summarize the conclusions in order to reflect only the study findings.

Reviewer 2 Report

We consider the study very interesting as it establishes the sensitivity and specificity of different radiological parameters that are useful for the diagnosis of flat feet. As a suggestion to the authors, some radiological images are missing in which the different anatomical references used to quantify the angles analyzed are included. Especially measurements with higher sensitivity and specificity such as CP angle and NI. 

On the other hand, and as far as possible, we recommend improving the quality of Figure 3.

Round 2

Reviewer 1 Report

In their first revision of manuscript, the authors have addressed my questions/comments properly.